# Substrateless Packaging for a D-Band MMIC Based on a Waveguide with a Glide-Symmetric EBG Hole Configuration

**DOI:** 10.3390/s22176696

**Published:** 2022-09-04

**Authors:** Weihua Yu, Abbas Vosoogh, Bowu Wang, Zhongxia Simon He

**Affiliations:** 1Beijing Key Laboratory of Millimeter-Wave and Terahertz Wave Technology, School of Integrated Circuits and Electronics, Beijing Institute of Technology, Beijing 100081, China; 2BIT Chongqing Institute of Microelectronics and Microsystems, Chongqing 400031, China; 3Microwave Electronics Laboratory, Department of Microtechnology and Nanosciense (MC2), Chalmers University of Technology, SE-41296 Gothenburg, Sweden; 4SinoWave AB, SE-43650 Hovås, Sweden

**Keywords:** electromagnetic band gap (EBG), mixer, packaging, MMIC to waveguide, transition

## Abstract

This paper presents a novel substrateless packaging solution for the D-band active e mixer MMIC module, using a waveguide line with a glide-symmetric periodic electromagnetic bandgap (EBG) hole configuration. The proposed packaging concept has the benefit of being able to control signal propagation behavior by using a cost-effective EBG hole configuration for millimeter-wave- and terahertz (THz)-frequency-band applications. Moreover, the mixer MMIC is connected to the proposed hollow rectangular waveguide line via a novel wire-bond wideband transition without using any intermediate substrate. A simple periodical nail structure is utilized to suppress the unwanted modes in the transition. Additionally, the presented solution does not impose any limitations on the chip’s dimensions or shape. The packaged mixer module shows a return loss lower than 10 dB for LO (70–85 GHz) and RF (150–170 GHz) ports, achieving a better performance than that of traditional waveguide transitions. The module could be used as a transmitter or receiver, and the conversion loss shows good agreement in multiple samples. The proposed packaging solution has the advantages of satisfactory frequency performance, broadband adaptability, low production costs, and excellent repeatability for millimeter-wave- and THz-band systems, which would facilitate the commercialization of millimeter-wave and THz products.

## 1. Introduction

Advances in semiconductor technologies have made the possibility of integrated circuits at millimeter-wave (mmW) and Terahertz (THz) frequency bands a reality. Such circuits have a wide range of applications, including communication, sensing, and imaging [1,2,3].

Driven by the growing demand for higher data rates, the operating frequency for wireless communication is moving towards to higher frequency bands. Today, the E band (71–76/81–86 GHz) is widely produced for point-to-point wireless links, allowing for multi-Gbps data rates; meanwhile, higher bands, from the W band to the H band, are in the research stage, providing a clear pathway for commercial development [4,5]. The free spectrum in the D band (130–134 GHz, 141–148.5 GHz, 151.5–164 GHz, and 167–174.8 GHz) is attractive because its available bandwidth is wider, and because it has relatively low atmospheric and rain attenuation. However, the commercialization of THz systems faces numerous challenges, such as cost-effective packaging techniques, the compact integration of active and passive components, and low-loss interconnectivity [6].

At frequencies above 100 GHz, substrat- based planar transmission line technologies show very high losses. Hollow metallic waveguides constitute a well characterized, low-loss transmission medium. The standard waveguide fabrication method is the so-called split-block, whereby the circuit structures are machined on two (or more) metal blocks and then joined together to form complete modules. An insertion loss of 0.2–0.25 dB/mm has been reported for such split-block standard WR-3 waveguide [7]. The split-block structure requires high quality fabrication by CNC milling and a precise assembly approach in order to obtain a good electrical/mechanical contact between the constitutive parts, which makes it very costly. When approaching THz frequencies, silicon micromachining offers a number of advantages for the fabrication of waveguide components, which becomes more beneficial [6,8]. However, the high cost of such components greatly limits the potential applications of THz technology, and it constitutes the primary bottleneck preventing its widespread application.

Gap waveguide (GWG) technology is a new approach intended to overcome the need for good electrical contact between the building blocks of a waveguide structure during mechanical assembly. It is based on the field cutoff obtained by two parallel perfect electric conductor (PEC) and perfect magnetic conductor (PMC) layers, which are separated by an air-gap smaller than a quarter wavelength. The PEC/PMC condition ensures the removal of any surface waves and parallel-plate modes within the air gap [9]. The PMC condition can be realized by an artificial magnetic conductor in the form of a periodic surface. Various passive components, such as antennas, band-pass filters, and diplexers, have been reported using the GWG technology [10,11,12,13]. Additionally, it is also important to integrate active components with GWG for its practical applications. In [14], a passive/active component gap waveguide transition interface for system integration working at the W band is proposed. In [15], a radio front-end at the E band, based on GWG technology for multi-Gbits/s backhaul links, has been demonstrated as a promising system packaging solution for frequencies from the mm-wave band to the THz band. Different fabrication methods, such as CNC milling, direct metal 3D-printing, and micromachining, are evaluated as means of fabricating these gap waveguide components [16,17,18]. However, the interconnection of active MMIC modules and packaging based on GWG technology are usually designed below 100 GHz, since the dimensions of the periodic structure are critical at higher frequency bands, and high fabrication costs become a drawback for the commercialization of GWG THz systems.

Glide-symmetric holey structures constitute an attractive alternative means of creating EBG surfaces at high frequencies. Such structures can provide the possibility of controlling signal propagation behavior in a cost-effective way [19,20]. The glide-symmetric configuration has advantages over pin-type EBGs; these include acquiring higher tolerance accuracy because of the larger unit cell periodicity, as well as a simpler and cheaper manufacturing process [21,22,23,24,25]. For example, a rectangular multilayer waveguide (MLW) concept with a glide-symmetric configuration has been proposed by stacking thin unconnected metal plates, without any requirement for electrical contact between the layers [21]. The EBG unit cells on each thin metal layer suppress the possible leakage of fields, which greatly simplifies the assembly process.

This paper presents a novel interconnection solution for packaging active components using glide-symmetric EBG structures at the D band. A mixer MMIC is connected to a hollow rectangular waveguide line via a substrateless transition. The proposed MMIC- to-waveguide wire-bond transition provides a wide bandwidth without the need for an intermediate substrate. The transition is implemented by bond-wire connections between the ground-signal-ground (GSG) on the MMIC and a U-shaped coupling slot on the broad wall of the waveguide. The periodical nails are utilized to suppress the propagation of unwanted modes at the top of the transition. The presented solutions do not impose any limitations on the chip’s dimensions or its shape. The packaged mixer module could be used as transmitter or receiver operating at the D band.

The paper is organized as follows. In Section 2, the process of designing the waveguide line with glide-symmetric EBG holes is presented. A detailed configuration and performance evaluation of the proposed transition are described in Section 3. In Section 4, the mixer module packaging process is proposed and discussed, including the measurement results of the proposed packaged module. Finally, some concluding remarks are given in Section 5.

## 2. Design of the Waveguide Based on a Glide-Symmetric Holey Configuration

As shown in Figure 1, a standard D-band rectangular waveguide (RW), made by two metallic plates, is considered to be the base waveguide groove. Such a groove is designed in the bottom layer, and the glide-symmetric holey EBG configuration is designed and located in the side walls of the groove to eliminate field leakage. Figure 2 shows the geometry of the glide-symmetric hole unit cell. A more detailed discussion of the design flow can be found in [19]. The holes in each layer have an offset of half of the period (*P*), with respect to those in the top and bottom layers. We choose a hole diameter versus period ratio (*d*/*P*) equal to 0.5 to obtain the maximum stopband bandwidth of the EBG unit cell, as explained in [19]. To evaluate the EBG unit cell performance, a reasonable air gap (*g*) of 0.01 mm between the two layers is considered. The dispersion diagram of this unit cell has been numerically calculated using the CST Eigenmode solver, as shown in Figure 3. The corresponding unit cell dimensions in the simulation are given in the caption of Figure 2. It can be seen that the unit cell provides a stopband over the frequency band from 110 GHz to 200 GHz. Thus, no electromagnetic modes can be propagated within the stopband.

A simulation model of the D-band air-filled GWG, based on the glide-symmetric holey EBG with two metal plates, is designed and shown in Figure 4. A GWG is formed by milling out an elongated 1.65 mm-wide channel in the metal block and stacking a thin metal layer on top of the channel.

After stacking the top and bottom layers, a rectangular GWG line with the dimensions 1.65 mm × 0.82 mm × 15 mm is formed; it is compatible with the standard WR-6.5 waveguide in the D band. Periodic rectangular slots along the groove channel are opened to decrease the wave travelling in the small gap between the two layers, as explained in [22]. CST Microwave Studio is used for the simulations. High-conductivity metal aluminum with dimensions of 3.56 × 10^7^ S/m is used during the simulation, and the simulated results are shown in Figure 5. With a gap distance from 0.01 mm to 0.03 mm, the relatively wide bandwidth shows good tolerance adaptivity with the designed EBG-based waveguide. When the gap is designed to be 0.01 mm, the reflection coefficients are below −28 dB in the whole D band, and below −40 dB for most of the band. The simulated transmission coefficient is better than −0.2 dB for almost all of the D band.

## 3. MMIC to GWG Transition Design

The GWG based on glide-symmetric EBG holes has H-plane slot cuts, which makes it suitable for the integration and packaging of active MMIC circuits into a compact waveguide system. For GWG application, the key component in active mmW/THz system packaging is the transition structure for MMICs. MMICs operating at such high frequencies usually rely on planar transmission lines, in particular the coplanar waveguide (CPW) or the microstrip (MS) line. Different methods and structures have been invented for realizing transitions between RW and CPW/MS with low losses and wide bandwidth requirements. The E-plane probe exhibits good performance, and various probe shapes have been investigated for the THz system [26,27,28]. Wideband on-chip antennas could eliminate the need for an external off-chip connection, and innovative packaging processes have been proposed, but their efficiency is low. A carrier substrate approach using a wire bonding probe transition shows good versatility [29,30,31,32,33]. However, the external probes usually need an aperture cut in the center of the broad wall of the waveguide line, which heightens the complexity of module fabrication and assembly. In addition, traditional bond wires entail high levels of series inductance and unwanted radiation, such that they suffer from poor repeatability and narrow bandwidth performance above 100 GHz. Integrating the waveguide transition on chip and coupling it directly to the waveguide is an attractive option, but it requires the co-design of the chip and the package, which is not available most of the time [34,35,36,37].

In this paper, a novel transition for chip-to-waveguide interconnection via direct gold wire bonding is proposed. In this approach, there is no need for any intermediate probe or transfer substrate between the chip and the GWG line. The proposed substrateless transition, operating at the D band, is illustrated in Figure 6. A GWG line is designed on the left side, and the other side is a GSG pad with a microstrip line on the GaAs substrate, which is compatible with most MMICs. In order to force the fields to propagate into the GWG and to suppress unwanted radiation at the transition, a simple periodical nail structure is designed and applied at the top of the transition. The inner two nails are shorter than the other nails, in order to leave enough space for the bond-wires. Two gold bond wires, using a JEDEC 4-point structure for better impedance matching, are used to connect the signal line of the MMIC to a ridge on the upper metal layer of the GWG.

The geometry parameters of the designed D-band transition are given in Table 1. As can be seen in Figure 7a, the transition is a quasi-planar structure and the electromagnetic wave is effectively guided into the GWG.

Figure 7a shows the electric field distribution of the transition from the view of the middle cutting plane. It can be seen that the transmitted signal couples to the waveguide via the bond wire. The unwanted radiation at the transition is minimized by using a periodical pin structure. The frequency responses of the transition with and without the periodical pin are shown in Figure 7b. Considering lossy aluminum with dimensions of 3.56 × 10^7^ S/m for the GWG line, and lossless GaAs for the MMIC, a return loss above 10 dB is obtained within the 134–167 GHz band, as well as a low insertion loss of 0.8 dB, including all losses from chip to the waveguide port.

Usually, CNC milling machines are used for the waveguide and leak proof cover. WEDM-LS are used to machine high-precision deep holes with small diameters for the upper layer’s EBG hole. Some parameter studies for the proposed transition (focused on 135–170 GHz for acceptable return loss) are presented in this section. This parameter study could be used for the design and sensitivity analysis of the transition. Figure 8 shows the effects of the slot width (*slot_W*) on the frequency response of the transition. It can be seen that the frequency response shifts towards lower frequencies when the width increases from 0.55 mm to 0.65 mm. The transition loss is below 1 dB and shows insusceptibility with the slot width. Figure 9 shows the effects of the slot length (*slot_L*) on the frequency response of the transition. It can be seen that the frequency response shifts towards lower frequencies when the length increases from 1.1 mm to 1.3 mm, while the 10 dB return-loss bandwidth remains basically unchanged. Using a high-precision wedge–wedge wire bonder, the minimum value of the wire arc height *H_w_* is 20 μm. Under ultra-low *H_w_* conditions, the distance between the two bonding pads D could be less than 150 μm. Thus, the minimum wire-bond wire length is about 187.5 μm. Considering small deviations for the bonding point, the length of the bond wire will be changed. Figure 10 depicts the effects of bond wire length on the transition. When the length changes from 250 μm to 300 μm, great sensitivity with this parameter is observed. A shorter bond wire would provide better return loss and better insertion loss. Precise dimensional control must be achieved with this parameter during the assembly.

Considering the mixer’s LO frequency of the E band, a transition with the same structure that was investigated at the D band has also been investigated at the E band. The design parameters are shown in Table 2, and the simulated results are shown in Figure 11. The transition has an insertion loss better than 0.2 dB, and a return loss above 15 dB for almost the whole E band.

## 4. Mixer-Packaged Module Configuration and Design

For the design of the D-band mixer packaging module, we choose a subharmonic mixer chip, with part number GMDR0035A from Gotmic AB. Typical performance values for this mixer MMIC are shown in Table 3. The chip is realized on the GaAs substrate and operates in the 145–170 GHz band for RF with 70–85 GHz for LO, and is suitable for D-band point-to-point communication, sensing, and imaging applications. A basic block diagram of the subharmonic mixer is shown in Figure 12. Figure 13 shows the bottom part of the design module, which houses the MMIC, the input/output waveguide channel, and the PCB that provides DC and IQ lines to the MMIC. LO/RF signals are transmitted in the EBG waveguides and connected with the mixer chip by direct wire bonding. Differential I/Q IF signals (0–6 GHz) are provided with coaxial interfaces, such as SMA connectors. For the PCB board, only the differential I/Q signal and the DC feed need to be traced, meaning that a low-cost substrate could be used. We have used the Rogers RO 4003C, which provides a stable dielectric constant and a low-loss tangent factor at the RF band. The PCB substrate thickness is 0.508 mm and it is gold plate afterwards, so that it is able to apply wire bonding. At operating frequency, the dielectric constant of this substrate is 3.55, its loss tangent is 0.0027, and the width of the 50 Ω microstrip line is about 1 mm. The top part acts as a metal box, with pin structures covering the LO and RF transitions.

**Figure 12 sensors-22-06696-f012:**
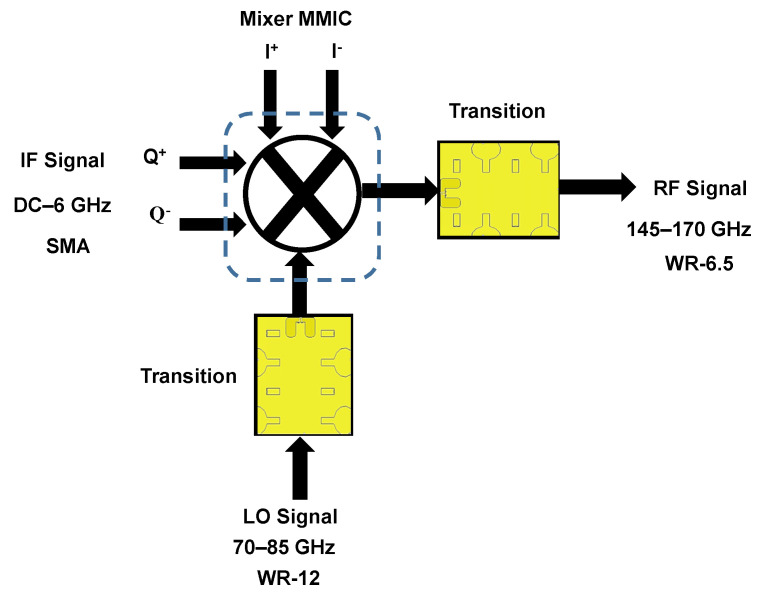
Block diagram of the IQ subharmonic up-mixer.

**Figure 13 sensors-22-06696-f013:**
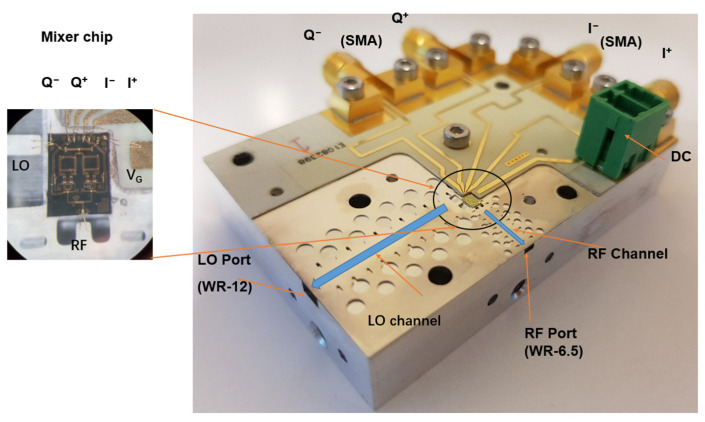
Fabricated D-band mixer assembly and detailed transition with the waveguide. To characterize the packaged mixer, measurements were performed using Keysight PNA-X N5247A and an E-band frequency extender for the LO port, while Keysight PNA-X and a D-band VDI frequency extender were used to measure the RF port separately. The LO/RF ports’ measured return loss for the mixer is shown in Figure 14; S11 is below −10 dB for most of the operating LO and RF working frequency bands, which is acceptable for the D-band mixer module. The simulated transition performance (ideal matching load port) and measured transition performances (real mixer port) show apparent differences, which are mainly caused by the mismatch between the transition and the mixer chip (GMDR0035A mixer presenting the port’s return loss of 10 dB).

**Figure 14 sensors-22-06696-f014:**
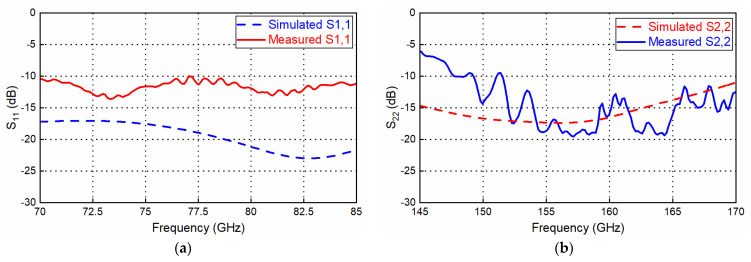
Return loss vs. frequency for the LO signal port (**a**) and the RF signal port (**b**) ① matching load.

In Table 4, the presented work is compared with the state-of-art packaging techniques reported in the literature. The comparison shows that the presented work exhibits the widest continuous working bandwidth with the active MMIC, and one of the lowest insertion losses; at the same time, it provides a substrateless packaging solution.

To measure the up-converter mode as a transmitter, the mixer measurement mode of Keysight PNA-X N5247A and a D-band VDI frequency extender were used. An IF signal with a quadrature differential input signal was generated by PNA-X through a quadrature coupler and two baluns. The LO signal was provided by an Agilent 8257D synthesizer together with an E-band ×6 multiplier module. The measured conversion loss and output power of the mixer with the up-converter mode as a transmitter are shown in Figure 15 and Figure 16, respectively. According to the GMDR0035A datasheet, the typical gate bias voltage (Vg) for the Mixer was −1.4 V, ranging from −1.7 V to −1.1 V. Three bias points were chosen to examine the optimum working conditions. IF is fixed at 2 GHz and LO is swept to cover RF at 145–170 GHz (upper side band). The fixed frequency response with swept IF power and several bias points is shown in Figure 17. With lower bias voltage, the mixer’s conversion loss is better, while output saturation comes earlier.

The same mixer module could also be used as a D-band receiver with a down-converter mode. When the RF input is settled with a power of −20 dBm, and the IF output is fixed at 2 GHz, the RF is changed from 145 GHz to 170 GHz, and the LO is changed to a USB configuration, and the result is shown in Figure 18. For the measurement of the S-parameters using a network analyzer, there is good agreement with the chip datasheet at most of the operating frequencies for both the up-converter mode and the down-converter mode. The mixer module has better CL flatness in the working frequency band compared with chip’s data, which is mainly caused by better wideband matching within the chip packaging.

Three modules were fabricated and measured. The conversion loss with the down-conversion mode is shown in Figure 19. There is almost no frequency shifting between the different modules, and the performance discrepancies are acceptable.

## 5. Conclusions

A GWG based on a glide-symmetric EBG hole configuration has the advantages of inexpensive fabrication, galvanic contact elimination, and leakage signal suppression, which makes the technology an excellent candidate for use in the millimeter and terahertz wave band. In this paper, a D-band IQ mixer MMIC module that uses this GWG configuration has been achieved, using a wire-bond transition from the MMIC chip to the GWG without any transfer substrate, resulting in a low-loss and highly efficient interconnection. Good agreement of the mixer module with the chip datasheet is achieved, which indicates that the GWG based on the glide-symmetric hole configuration, and the substrateless transition used, have good frequency performances at the D band. The good consistency between multiple modules shows that the method developed here is highly repeatable and suitable for large-scale fabrication.

We have demonstrated that the implementation of mmW/THz active devices is feasible using the proposed integration and packaging solution. Actually, the same method could be used to construct a whole compact waveguide transceiver system with low costs and high performance, which makes the commercialization of mmW/THz applications more feasible.

## Figures and Tables

**Figure 1 sensors-22-06696-f001:**
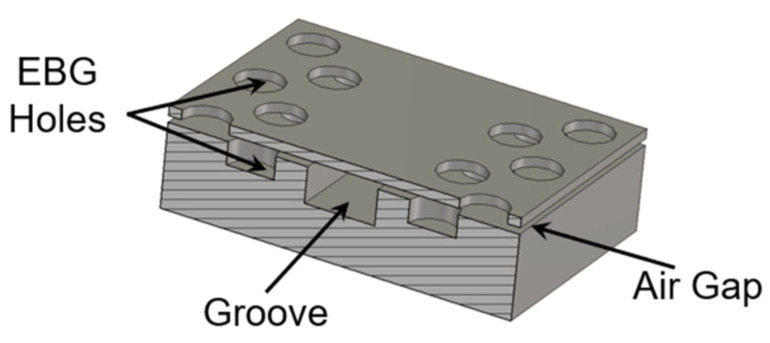
Configuration of the EBG rectangular waveguide line using glide-symmetric holes.

**Figure 2 sensors-22-06696-f002:**
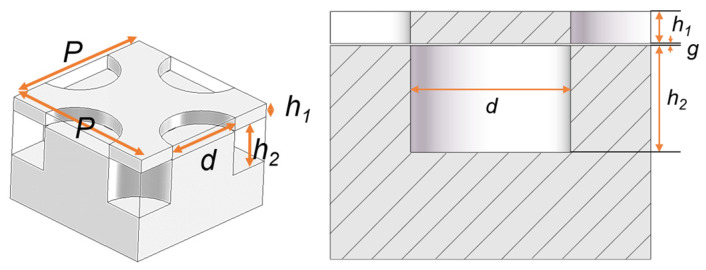
Configuration of the unit cell with glide-symmetric holes, consisting of two metal layers. (*P* = 2 mm, *d* = 1 mm, *h*_1_ = 0.15 mm, *h*_2_ = 0.5 mm, *g* = 0.01 mm).

**Figure 3 sensors-22-06696-f003:**
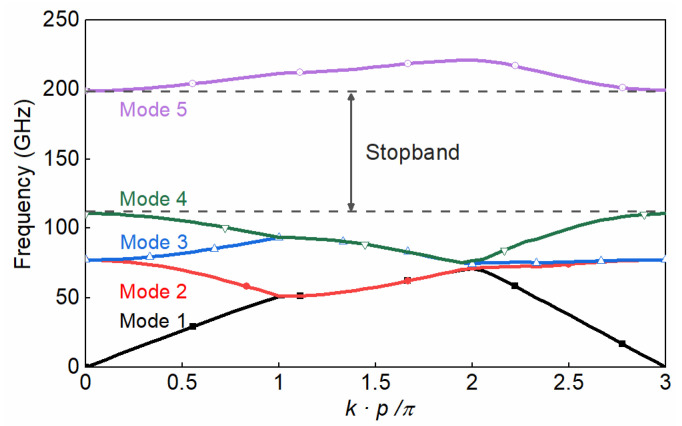
Dispersion diagram for the infinite periodic unit cell.

**Figure 4 sensors-22-06696-f004:**
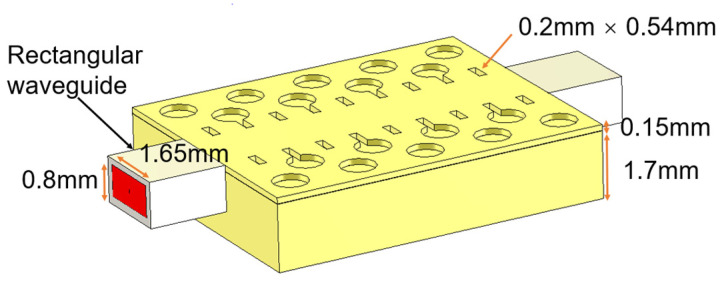
Simulation model of the air-filled GWG, based on the glide-symmetric holey EBG.

**Figure 5 sensors-22-06696-f005:**
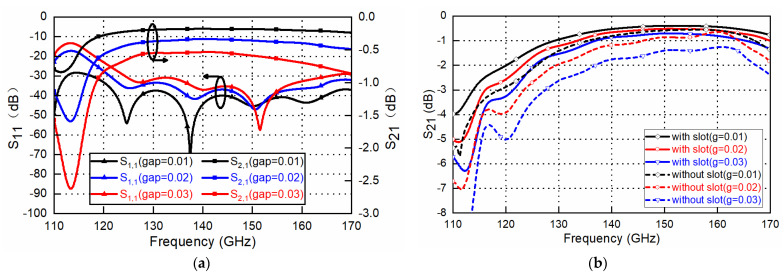
Simulated frequency response of the designed D-band GWG line (**a**) and with and without rectangular slots along the groove channel (**b**).

**Figure 6 sensors-22-06696-f006:**
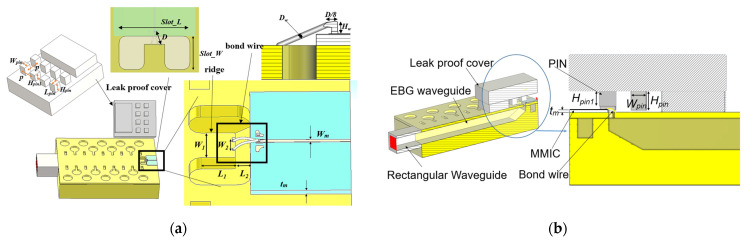
Proposed chip-to-waveguide transition: (**a**) the top-down view of the transition and the associated important dimension parameters; (**b**) the perspective view, split in half.

**Figure 7 sensors-22-06696-f007:**
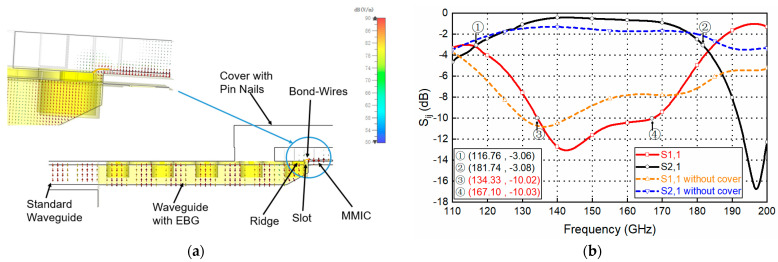
Substrateless transition. (**a**) E-field distribution of the transition; (**b**) simulated performances of the transition.

**Figure 8 sensors-22-06696-f008:**
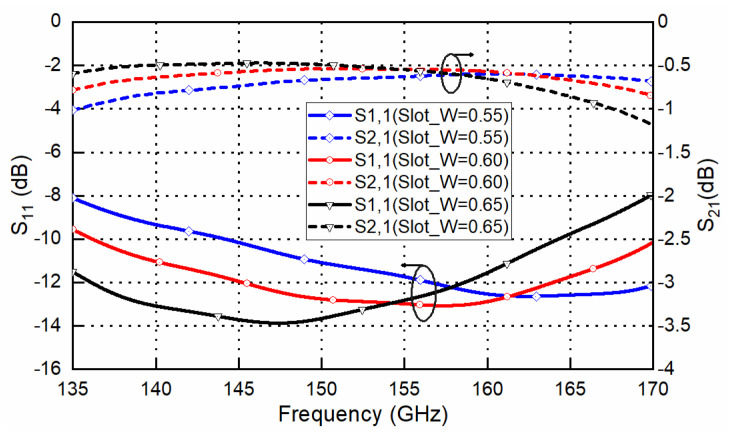
Effect of the slot width (slot_W) on the designed transition frequency response.

**Figure 9 sensors-22-06696-f009:**
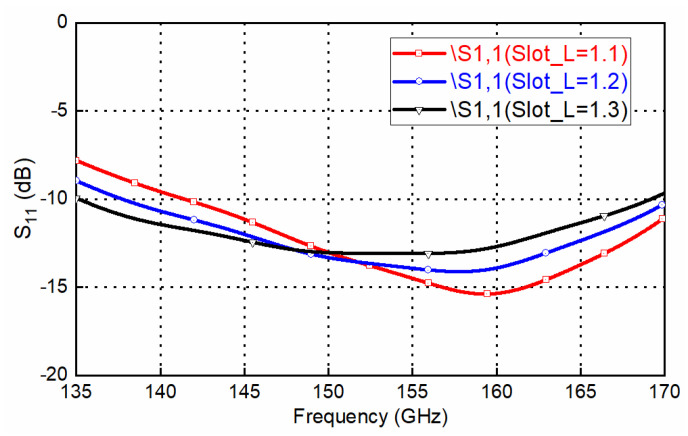
Effect of the slot length (slot_L) on the designed transition frequency response.

**Figure 10 sensors-22-06696-f010:**
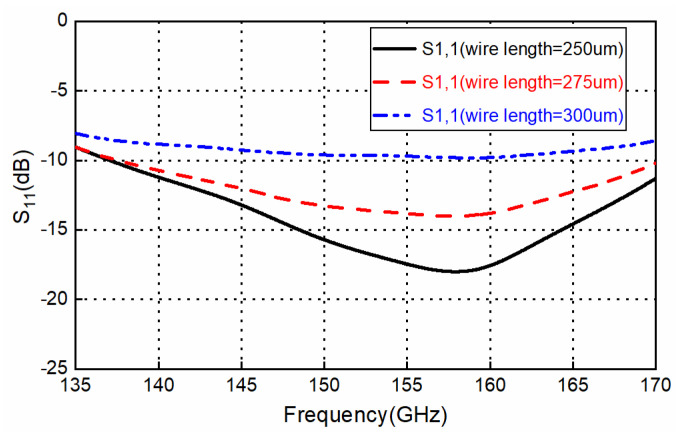
Effect of the bond wire length on the designed transition frequency response.

**Figure 11 sensors-22-06696-f011:**
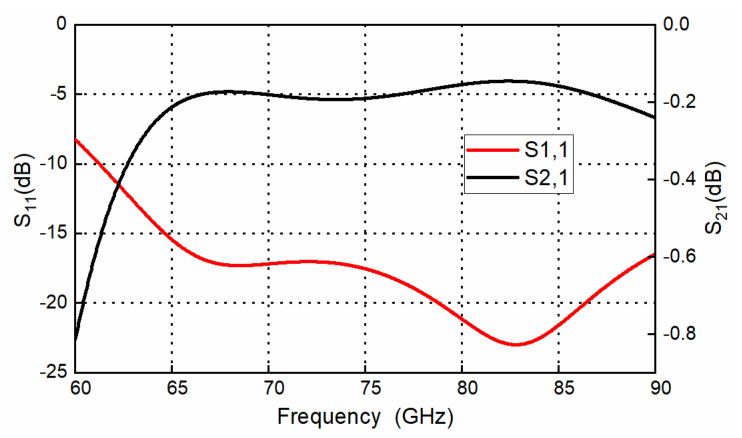
Simulated frequency response of the designed E-band transition.

**Figure 15 sensors-22-06696-f015:**
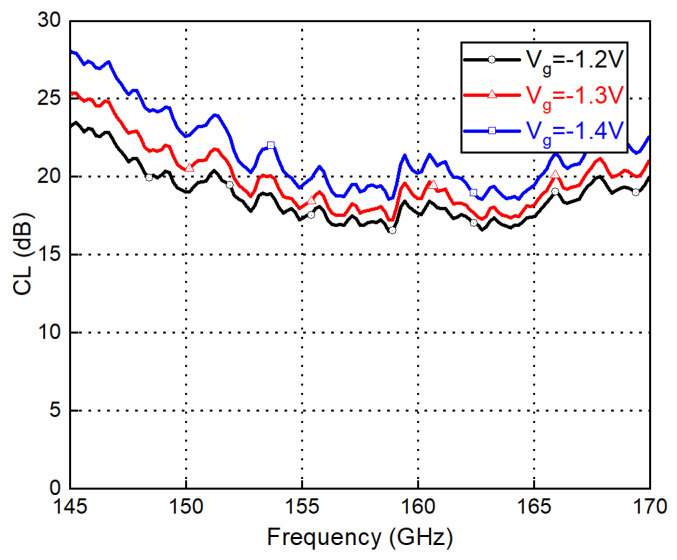
Conversion loss (CL) vs. frequency (IF @ 10 dBm and LO @ 13 dBm).

**Figure 16 sensors-22-06696-f016:**
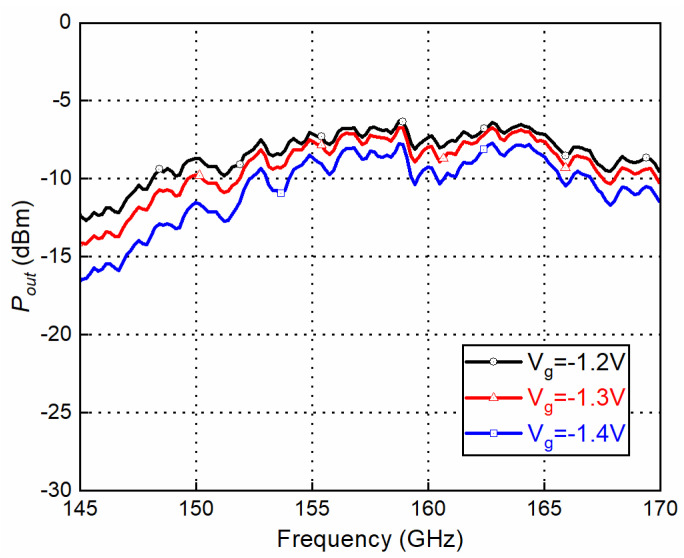
Output power vs. frequency (IF @ 10 dBm and LO @ 13 dBm).

**Figure 17 sensors-22-06696-f017:**
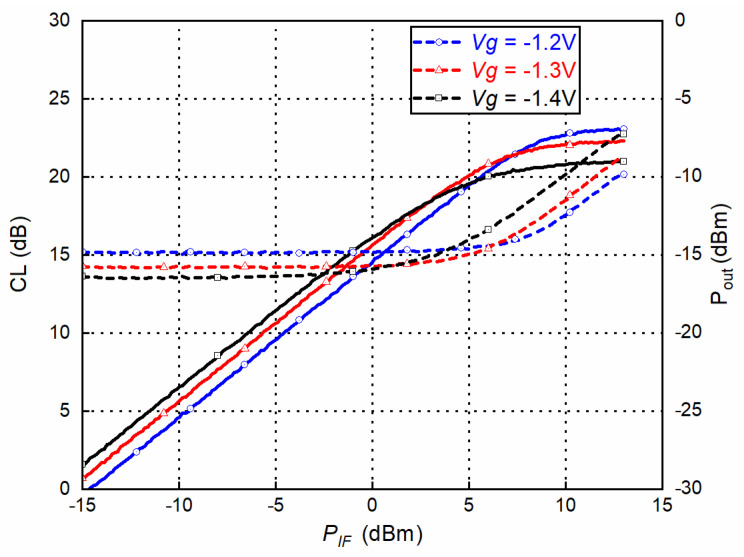
Conversion loss (CL) and output power vs. input power at 160 GHz. (LO @ 13 dBm).

**Figure 18 sensors-22-06696-f018:**
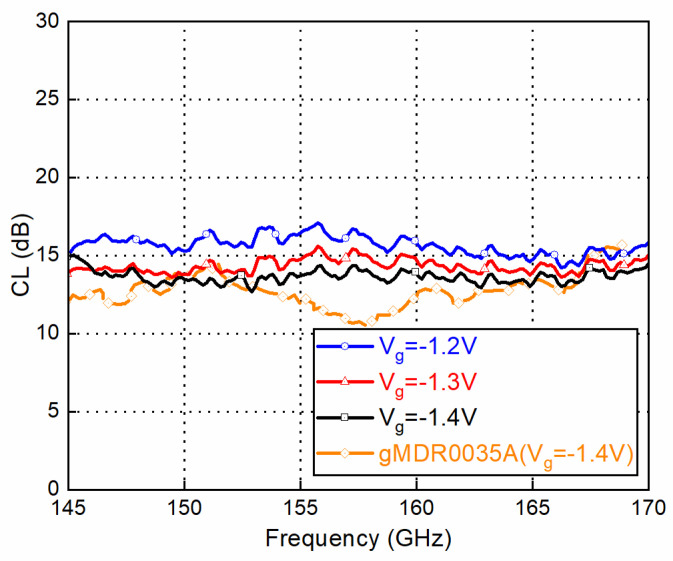
Conversion loss (CL) vs. frequency. (RF @ −20 dBm and LO @ 13 dBm).

**Figure 19 sensors-22-06696-f019:**
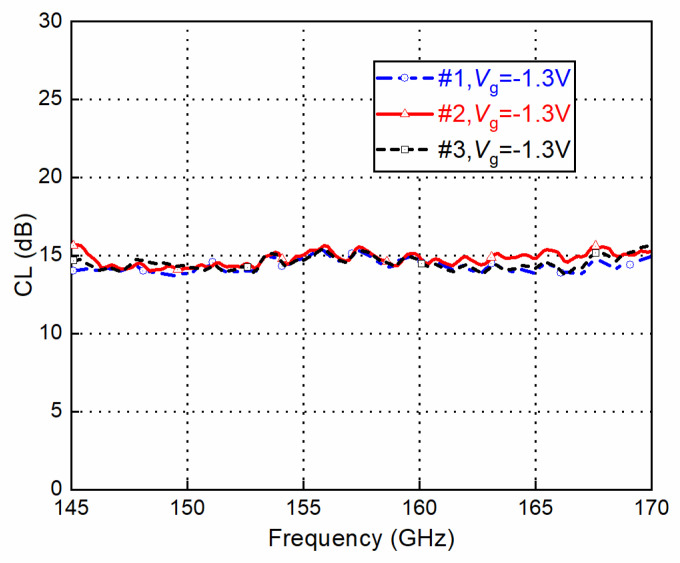
Conversion loss (CL) vs. frequency. Comparison between the three modules.

**Table 1 sensors-22-06696-t001:** Optimum geometry parameters for the MMIC-to-GWG transition, based on the glide-symmetric holey EBG.

Parameter	Value (mm)	Parameter	Value (mm)
*W_1_*	0.373	*W_m_*	0.035
*W_2_*	0.1	*t_m_*	0.05
*L_1_*	0.34	*W_pin_*	0.4
*L_2_*	0.15	*L_pin_*	0.4
*D_w_*	0.025	*H_pin_*	0.5
*H_w_*	0.03	*H_pin1_*	0.4
*D*	0.2	*p*	0.8
*Slot_W*	0.6	*Slot L*	1.3

**Table 2 sensors-22-06696-t002:** Optimum dimension parameters for the MMIC-to-waveguide transition, based on a glide-symmetric holey EBG, at the E band.

Parameter	Value (mm)	Parameter	Value (mm)
*W_1_*	0.5	*W_m_*	0.035
*W_2_*	0.18	*t_m_*	0.05
*L_1_*	0.565	*W_pin_*	0.5
*L_2_*	0.15	*L_pin_*	0.5
*D_w_*	0.025	*H_pin_*	0.9
*H_w_*	0.03	*H_pin1_*	0.8
*D*	0.35	*p*	1
*Slot_W*	1.1	*Slot L*	2.9

**Table 3 sensors-22-06696-t003:** Typical performance values of the GMDR0035A chip.

Parameter	Min	Typ	Max	Unit
RF frequency	140		170	GHz
IF frequency	DC		6	GHz
LO frequency	70		85	GHz
LO input power		15		dBm
LO multiplication factor		2		
Conversion loss		12		dB
P1dB		−5		dBm
RF return loss	10			dB
IF return loss	10			dB
LO return loss	10			dB

**Table 4 sensors-22-06696-t004:** Measurement comparison of D-band transitions.

No.	Process	Measured Topology	Bandwidth	Freq Band/GHz ^a^	Insertion Loss/dB ^b^	Return Loss/dB ^c^
[6]	SiGe BiCMOS with heterogeneous integration	Transition + antenna	16.95%	135–160	4.2–5.5	>5
[33]	127 μm-thick quartz with bonding-wire interconnect	Transition with single-ended ML	16.4%	144.7–170	2	10
[36]	SiC chip with nongalvanic packaging	Transition with back-to-back ML	23.7%	125–137 147–170	0.7	~12
[37]	eWLB with silicon taper interconnect	Transition with single-ended ML	26%	116–151	3.4	5
This work	GaAs chip with substrateless packaging	Transition with single-ended mixer	21.92%	134–167	0.8	>10

^a^ working frequency band with return loss better than 10 dB. ^b^ Average insertion loss. ^c^ Maximum return loss.

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
