# Peer review of "Substrateless Packaging for a D-Band MMIC Based on a Waveguide with a Glide-Symmetric EBG Hole Configuration"

_sensors, 2022, doi:10.3390/s22176696_

Round 1

Reviewer 1 Report

(1) The measurement setup is suggested to be given in the text.

(2) The details of Fig.13 should be provided. Besides, the photo of the mixer MMIC is not clear.

(3) The curves' qualities should be improved in the manuscript.

(4) The number of the references in Table3 should be organized.

(5) Could the results of the co-simulation of circuit simulation using SNP files and the full-wave simulation be provided in Fig. 14 for comparison?

(6) The datasheet performances of the GMDR0035A from gotmic AB are suggested to be summarized in a Table in the manuscript.

Author Response

Reply to the Reviewer 1’s Comments

  • Point 1:The measurement setup is suggested to be given in the text.

Response:We have added measurement setup instructions to the manuscript in lines 264 to 266 of the manuscript, ‘Measurement has been performed to characterize the packaged mixer with Keysight PNA-X N5247A and a E-band frequency extender for LO port, while with Keysight PNA-X and D-band VDI frequency extender for RF port separately.’

And in lines 285 to 288 of the manuscript, ‘To measure the up-converter mode as a transmitter, mixer measurement mode of Keysight PNA-X N5247A and a D-band VDI frequency extender were used. IF signal with the quadrature differential input signal was generated by PNA-X through a quadrature coupler and two baluns.’ .

  • Point 2:The details of Fig.13 should be provided. Besides, the photo of the mixer MMIC is not clear.

Response:We've provided clearer images for Fig. 13 and updated image descriptions in lines 241 to 253 of the manuscript. ‘Differential I/Q IF signals (0-6 GHz)are provided with coaxial interfaces like SMA connectors. For on the PCB board, only differential I/Q signal and DC feed need to be traced, a low-cost substrate could be used. We have used the Rogers RO 4003C that provides a stable dielectric constant and low-loss tangent factor at RF band. The PCB substrate thickness is 0.508 mm and is gold plate afterward to be able to apply wire-bonding. At operating frequency, the dielectric constant of this substrate is 3.55, its loss tangent is 0.0027, and the 50Ω microstrip line width is about 1 mm. The top part acts as a metal box with pin structures covering the LO and RF transitions..’

  • Point 3:The curves' qualities should be improved in the manuscript.

Response:We have re-updated the format and resolution of the curve images.

  • Point 4:The number of the references in Table3 should be organized.

Response : We have reorganized The number of the references in Table3, and we have renamed it Table4.

Table 4. Measurements Comparison of D-band Transition.

No.

Process

Measured Topology

Bandwidth

Freq Band/GHza

Insertion Loss/dBb

Return Loss/dBc

[6]

SiGe BiCMOS with Heterogeneous Integration

Transition+Antenna

16.95%

135-160

4.2-5.5

>5

[36]

SiC chip with

Nongalvanic packaging

Transition with ML

Back-to-Back

23.7%

125-137

147-170

0.7

~12

[37]

127 μm-thick quartz with Bonding-wire Interconnect

Transition with ML

Single-ended

16.4%

144.7-170

2

10

[38]

eWLB with Silicon Taper Interconnect

Transition with ML

Single-ended

26%

116-151

3.4

5

This work

GaAs chip with

Substrate-Less Packaging

Transition with Mixer

Single-ended

21.92%

134-167

0.8

>10

  • Point 5:Could the results of the co-simulation of circuit simulation using SNP files and the full-wave simulation be provided in Fig. 14 for comparison?

Response: For the reason of the GMDR0035A datasheet only give the electrical performance       for RF/LO return loss is 10 dB, no SNP files could be provided, therefore when designing the   transition from MMIC to the waveguide, an ideal matching load was chosen to make the best of  it. And with the mixer chip, the measured return loss show the effects with non-ideal mmic with 10 dB or so is reasonable.

  • Point 6:The datasheet performances of the GMDR0035A from Gotmic AB are suggested to be summarized in a Table in the manuscript.

     Response: In the new Table3, we provide the datasheet performances of the GMDR0035A from gotmic AB.

Table 3. Typical performance values of chip GMDR0035A.

Parameter

Min

Typ

Max

Unit

RF frequency

140

170

GHz

IF frequency

DC

6

GHz

LO frequency

70

85

GHz

LO input power

15

dBm

LO multiplication factor

2

Conversion loss

12

dB

P1dB

-5

dBm

RF return loss

10

dB

IF return loss

10

dB

LO return loss

10

dB

Reviewer 2 Report

file attached

Author Response

  • Point 1:The authors should improve the use of the English language in the manuscript text and the figures, and define abbreviations when used for the first time.

Response: We have re-edited the figures and improved English language usage. And the abbreviations are defined the first time they are used.